SCIENCE FORUM

# Antibody characterization is critical to enhance reproducibility in biomedical research

**Abstract** Antibodies are used in many areas of biomedical and clinical research, but many of these antibodies have not been adequately characterized, which casts doubt on the results reported in many scientific papers. This problem is compounded by a lack of suitable control experiments in many studies. In this article we review the history of the 'antibody characterization crisis', and we document efforts and initiatives to address the problem, notably for antibodies that target human proteins. We also present recommendations for a range of stakeholders – researchers, universities, journals, antibody vendors and repositories, scientific societies and funders – to increase the reproducibility of studies that rely on antibodies.

**RICHARD A KAHN\*, HARVINDER VIRK, CARL LAFLAMME, DOUGLAS W HOUSTON, NICOLE K POLINSKI, ROB MEIJERS, ALLAN I LEVEY, CLIFFORD B SAPER, TIMOTHY M ERRINGTON, RACHEL E TURN, ANITA BANDROWSKI, JAMES S TRIMMER, MEGHAN REGO, LEONARD P FREEDMAN, FORTUNATO FERRARA, ANDREW RM BRADBURY, HANNAH CABLE, SKYE LONGWORTH**

**\*For correspondence:** rkahn@emory.edu

## Introduction

Antibodies are critical reagents used in a variety of assays and protocols in biomedical and clinical research. The ability to detect, quantify, enrich, localize, and/or perturb the function of a target protein – even when present in a complex protein mixture, such as a cell lysate or tissue slice, or even in an intact organism – is key to many biomedical research studies. Similarly, the ability to detect changes in protein levels, localization, or interactions with other proteins or membranes, is critical when seeking to identify the pathways involved in cell regulation and disease pathologies.

For these reasons, the market for antibodies has soared, growing from ~10,000 commercially available antibodies about 15 years ago, to more than six million today (*Longworth and Chalmers, 2022*). However, it has been estimated that ~50% of commercial antibodies fail to meet even basic standards for characterization, and this problem is thought to result in financial losses of $0.4–1.8 billion per year in the United States alone (*Ayoubi et al., 2023*; *Bradbury and Plückthun, 2015*; *Voskuil et al., 2020*; *Baker, 2015b*). Moreover, the problems caused by the variable quality and characterization of commercial antibodies are compounded by end users not receiving sufficient training in the identification and use of suitable antibodies.

Together these issues have led to an alarming increase in the number of scientific publications that contain misleading or incorrect interpretations and conclusions because they are based on data from experiments that used antibodies that had not been properly characterized or validated (*Goodman, 2018*; *Menke et al., 2020*; *Laflamme et al., 2019*; *Aponte Santiago et al., 2023*). This situation, and the resulting problems with reproducibility, has been termed a 'crisis' (see, for example, *Baker, 2015b*). There is also a growing body of data that includes stark demonstrations of the volume of incorrect or misleading

data published, including clinical patient trials (see, for example, *Andersson et al., 2017*, *Virk et al., 2019*, and table 1 in *Voskuil et al., 2020*), based upon the use of poorly characterized antibodies. (It should be noted, however, that therapeutic antibodies – unlike research antibodies – are very well regulated and are subject to strict controls involving manufacturer and clinical trials; *Longworth and Chalmers, 2022*).

The roots of this now pervasive problem can be traced back to the early 2000 s, when the first near-complete human genome sequence became available, and attention turned to using this information to make further discoveries in biomedical sciences. Analyses of the genome resulted in open questions as to how many proteins are encoded and which are expressed in different tissues. Such discussions resulted in formation of the Human Proteome Organization (*Omenn, 2021*) and its Human Proteome Project (*Legrain et al., 2011*). As suggested by the name, the Human Proteome Project focused on the determination of the human proteome, based upon experimental evidence, and the development of tools (notably mass spectrometry and antibodies) to enable further research. They also developed three foundational approaches to determine the human proteome and make it available to researchers: (i) shotgun and targeted mass spectrometry; (ii) polyclonal and monoclonal antibodies; (iii) an integrated database, intended to promote sharing of data and a platform for multiple uses.

The Human Protein Atlas was launched around the same time and had similar goals: to map all human proteins in cells, tissues, and organs. This project's approach was highly dependent upon use of antibodies in immunohistochemistry and immunofluorescence (*Berglund et al., 2008*). Thus, as research efforts turned from the genome to the proteome, there was a clear need for a large increase in the number of antibodies with the requisite affinities and specificities to meet the needs of these initiatives. However, over the following 20 years, concern over the lack of consensus for how best to validate antibody usage grew and, perhaps worse, it became clear that many end users did not adequately understand how the quality of their data depended on the antibodies they used being properly validated (see, for example, *Gloriam et al., 2010*; *Bandrowski, 2022*; *Singh Chawla, 2015*; *Blow, 2017*; *Lund-Johansen, 2023*; *Couchman, 2009*; *Williams, 2018*; *Pillai-Kastoori et al., 2020*; *Andersson et al., 2017*; *Sivertsson et al., 2020*;

*Polakiewicz, 2015*; *Howat et al., 2014*; *Edfors et al., 2018*; *Gilda et al., 2015*).

In the early days of antibody research most antibodies were generated and characterized in research labs, and sent to other research groups upon request. But as demand increased, companies selling antibodies to researchers became increasingly important. At first these companies relied on researchers supplying them with antibodies that had already been characterized, and the researchers received some fraction of the sale price in return.

Later, many companies started to generate the antibodies themselves, particularly for the most widely used antibodies, as these were the most profitable. It is important for anyone hoping to understand and address the antibody crisis to realise that the costs associated with performing even the most basic/common characterizations (Western blotting, immunoprecipitation, immunofluorescence and immunohistochemistry), and obtaining appropriate knockout cell lines, is many times higher than the revenues generated from the average antibody on the market today. Thus, the current system puts the onus on end users to both find the best antibody on the market for them, and to perform appropriate characterization prior to using the antibody – otherwise they could waste a lot of time and money on experiments that do not produce meaningful or trustworthy results. Finding the best antibody for your research, and then validating it, can be quite challenging, so below we describe some resources that can help.

A related issue is that with the focus placed on the most widely used antibodies, the research community is failing to take advantage of all the information contained in the human genome sequence because the high-quality reagents (including antibodies) needed to investigate a large fraction of the proteome are simply not available or identifiable as a result of the lack of characterization data linked to them (*Edwards et al., 2011*).

The origins of this antibody crisis are many and varied, so it is unrealistic to expect a complete 'solution' in the near future. However, there are a number of steps can be taken to dramatically improve the situation, such as: continuing to raise awareness of the issues among end users; more accurately identifying the reagents on vendor websites and in scientific publications; and sharing characterization data, when available.

In this article we assemble a history of key events in the field (*Figure 1*); summarize efforts to raise awareness of the problem – and ways to

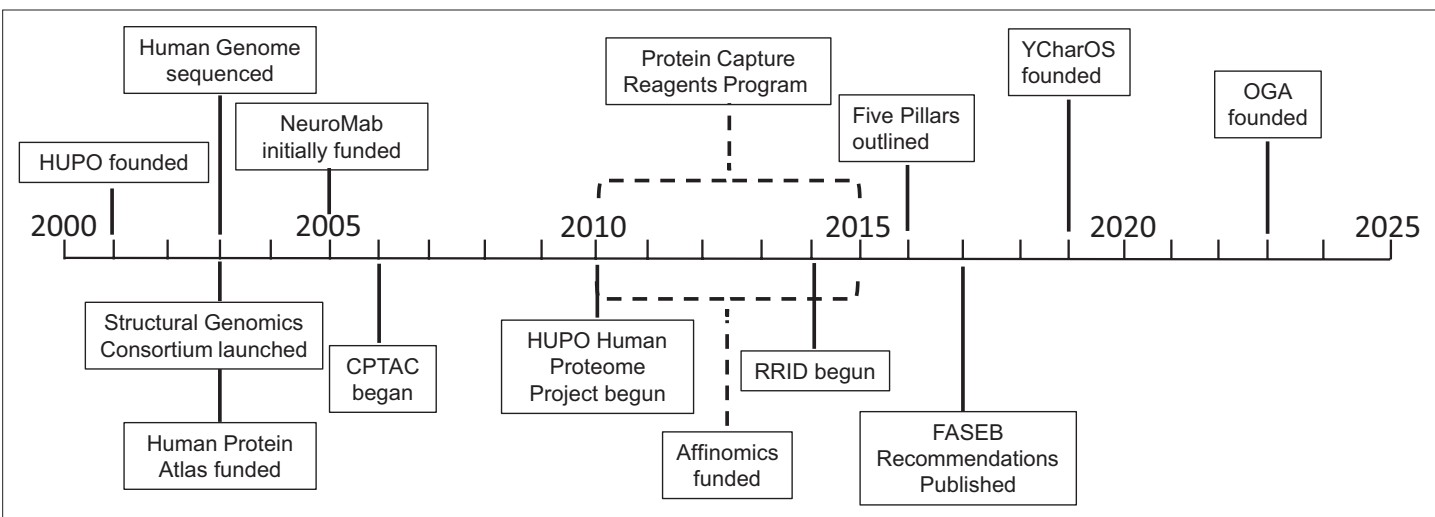

**Figure 1.** Key projects related to antibody characterization. Timeline showing when the projects discussed in this article were started: the Protein Capture Reagents Program and the Affinomics project were funded for five years (as indicated by dashed lines). CPTAC: Clinical Proteomic Tumor Analysis Consortium; FASEB: Federation of American Societies of Experimental Biology; HUPO: Human Proteome Organization; OGA: Only Good Antibodies; RRID: Research Resource Identifier; YCharOS: Antibody Characterization through Open Science.

address the problem – through editorials, meetings, and workshops; and discuss a number of initiatives that are trying to address the problem. The role of technology development in the field, and its role in planning future initiatives, is also discussed. In the last section we propose actions for the various stakeholders – researchers and end users, universities, journals, antibody vendors and repositories, scientific societies, charities and funding agencies – that we believe will help address the antibody characterization and its huge, ongoing and detrimental impact on reproducibility. Clearly, a consensus amongst all stakeholders is required to resolve the antibody characterization crisis.

## Terminology

The term antibody 'characterization' is more apt than 'validation', as an antibody's suitability often varies across different assays. The term antibody 'characterization' is best applied to a description of the inherent ability of an antibody to perform in different assays – i.e. for a recombinant antibody it describes the properties of an antibody with a specific sequence (e.g. functional in Western blot, immunohistochemistry and immunofluorescence, but not in immunoprecipitation). While validation is best applied to confirmation that a particular antibody lot received in a lab performs as characterized – an antibody may not be validated if (for example) it has been inadvertently denatured by environmental extremes.

Antibody characterization can be thought of as simply the controls needed in any protocol that uses an antibody, to ensure that the antibody is performing with the requisite specificity needed to ensure accurate interpretation of the results. In contrast, the term validation is more appropriate when demonstrating context-specific suitability for an application. Also, the terms 'specificity' and 'selectivity' are often used interchangeably, though the former refers to the ability to bind only to its intended target, while the latter refers to its ability to bind that protein even when in a complex mixture of proteins. For a concise introduction to the use of antibodies in research, particularly in immunohistochemistry, including the basic principles of antibody action and the different types of antibodies, see *Saper, 2009*.

Readers should also be aware that some terms in the field are used interchangeably, even though they mean slightly different things, which can be confusing. For example, immunofluorescence, immunohistochemistry, and immunocytochemistry are all used to localize targets in cells or tissues, which may involve the use of a variety of different fixatives or permeabilization steps. However, the key differences are in the method of detection: the term immunofluorescence should be used when fluorescence is the output, while the other terms are more appropriate when the readout is generated by enzymatic labels, such as peroxidase and alkaline phosphatase. Also, immunofluorescence can be used on live cells, while the other two approaches require fixation.

In this article we will discuss just immunofluorescence and immunohistochemistry for the sake of simplicity.

Complexity arises from the use of antibodies across different experimental protocols, each requiring distinct controls. Common applications include Western blotting (also known as immunoblotting), immunoprecipitation, immunofluorescence on cultured cells, immunohistochemistry on tissue sections, flow cytometry with intact cells in suspension, enzyme-linked immunosorbent assay (ELISA), and functional assays. Characterization of an antibody should include testing in as many of these assays as feasible to determine its potential uses and value to researchers. However, the combination of studies needed to validate an antibody for a particular use depends upon the experiment that is being undertaken. In other words, characterization data from vendors, publications and public databases can be helpful when identifying the best candidate antibodies for a specific use, but researchers should always confirm that the antibodies they plan to use will perform as needed in their experiments.

The prevalent use of polyclonal antibodies, derived from immunized animals and most often used in the form of serum from such animals, is a large source of the problems described herein. This is due to their non-renewable nature and the complexity of the different antibodies present, which can influence batch variability as a result of the presence of both specific and non-specific antibodies. This may introduce false positives and increased background noise, and is further confounded when serum from different bleeds or animals is sold under the same name and catalog number. Also, the profile of a polyclonal antibody response can vary over time, even with affinity purification, as the antibody population in each batch is varied. Despite these drawbacks, polyclonal antibodies remain widely used, necessitating a coordinated characterization effort.

Monoclonal antibodies have historically been generated by fusion of an immortal myeloma cell line with terminally differentiated B cells from an immunized animal. Cloning of the resulting hybridoma cells, that each secrete a single antibody, allows one to identify antibodies that bind the target protein in different assays and to generate large batches of that single antibody. Thus, monoclonal antibodies were the first type of renewable antibody as they could be generated repeatedly and would remain consistent. However, caution is advised as hybridoma lines can vary over time and may express more than one antibody (*Bradbury et al., 2018*; *Mitchell et al., 2023*).

Recombinant antibodies represent a newer technology, involving the determination of the DNA sequence encoding the antigen binding site. This allows cloning it into plasmids to allow expression of a single antibody, offering stable and renewable reagents, with customizable constant regions for varied applications and much more flexibility in use. Newer technologies allow culture of single B cells from an immunized animal, allowing cloning of the secreted antibody to generate a recombinant antibody. The use of large libraries that encode many millions of potential antigen binding regions, displayed on the surface of phage particles or yeast cells, has allowed high-throughput screening for high affinity reagents that have the added value of not requiring the use of any animals in their development. Some researchers believe that an animal's immune system may still do a better job of identifying the best epitopes for targeting than even a large library of phage or yeast cells, though this remains somewhat controversial (*Custers and Steyaert, 2020*).

Herein, we focus on antibodies that target proteins (as opposed to polysaccharides, nucleic acids or other cellular components) and use the term epitope to indicate the specific sites/residues within a given target protein (antigen) to which an antibody binds. It is worth noting here that the same epitope can be present on other proteins. The ideal antibody would bind to a unique, linear epitope that is fully exposed on the surface of the folded protein and not involved in any post-translational modifications or protein–protein interactions, allowing equal antibody access in both native and unfolded conditions and without regard to tissue/cell-specific differences in protein associations or post-translational modifications. In contrast, there are many examples of very useful antibodies that bind the antigen in a conformation dependent fashion, and thus fail to bind to the denatured protein (in, for example, Western blotting). Similarly, antibodies that bind to epitopes that include specific post-translational modifications can be valuable for monitoring such modifications over time and space. While potentially valuable reagents, such antibodies present challenges in characterization that are not discussed here. The affinity of any antibody should be low nanomolar or tighter, to allow detection of low abundance, endogenous protein, while maintaining high specificity for a single protein.

## How awareness of the antibody crisis grew

In the early 2000s, researchers and journals began raising the alarm about the growing antibody crisis, explaining why appropriate controls and detailed/accurate reporting of antibodies was so important (*Saper, 2005*; *Saper, 2009*; *Saper and Sawchenko, 2003*). In a review article titled 'Antibody Validation' that was published in 2010, Bordeaux and co-authors discussed the importance of antibody characterization in some detail (*Bordeaux et al., 2010*), while also stressing the need to differentiate between the lack of antibody characterization and scientific misconduct or data manipulation (*Collins and Tabak, 2014*). This article also underscored common pitfalls in antibody use, including a lack of appropriate user training, over-reliance on vendors for characterization, inadequate methodological details from providers and in publications, and challenges in reagent identification. Bordeaux et al. also highlighted the problems that emerge when antibodies are used without characterization, and cited numerous examples of such misuse.

While we agree with the statement that 'the responsibility for proof of specificity is with the purchaser, not the vendor' (*Bordeaux et al., 2010*), we argue herein that all stakeholders have responsibilities when it comes to addressing the antibody crisis. Moreover, we encourage the use of knockout (KO) or knockdown (KD) cell lines or tissue samples as important negative controls for specificity. KO cell lines and model organisms have become much more readily available, thanks to CRISPR technologies, and the use of antibodies to confirm KO of a protein is just as useful a tool as the use of the KO lines to test for specificity of antibodies. Unfortunately, there is currently no repository that researchers can use to share KO cell lines. As useful as KO cells/tissues are as negative controls for specificity, the characterization of antibodies is always further improved when combined with other approaches.

A number of articles in various *Nature* journals contributed to the conversation about the antibody crisis, including an editorial (*Methods, 2015*), various news articles (*Baker, 2015b*; *Baker, 2015a*; *Baker, 2016*; *Baker, 2020*), a comment article about reproducibility issues in preclinical cancer research (*Begley and Ellis, 2012*), and another comment article with over 100 co-signatories that asked the National Institutes of Health and European Union "to convene academic users, technology developers, biotech companies, funding agencies, and publishers,

and establish a realistic timetable for the transition to these [recombinant antibodies] high quality binding reagents" (*Bradbury and Plückthun, 2015*).

The Global Biological Standards Institute (GBSI), established in 2012, played a significant role by forming a Research Antibodies and Standards Task Force, and its 2015 online survey and subsequent analyses (*Freedman et al., 2016*) emphasized the urgent need for better training for end users and for standards for best practices. Another GBSI study found that the US spends ~$28 billion per year on preclinical research that is not reproducible, and concluded that urgent improvements were required in two areas – study design and the validation of biological reagents (*Freedman et al., 2015*). The GBSI has also emphasized the need for the validation of cell lines, not just antibodies (*Souren et al., 2022*; *Marx, 2014*). A GBSI meeting in Asilomar in 2016 focused on developing antibody characterization standards (*Baker, 2016*), and although the scoring system proposed at this meeting has seen limited use, webinars developed in concert with the Antibody Society *Voskuil et al., 2020* have proven valuable in educating users.

The Federation of American Societies of Experimental Biology (FASEB) also published an important report called *Enhancing Research Reproducibility* in 2016, which stressed the need for standard reporting formats for antibodies (*FASEB, 2016*). More recently, in 2023, the annual meeting of the American Society for Cell Biology included a workshop on antibody characterization. One conclusion from this workshop was that venders and researchers "need to adopt a standard format for reporting antibody information."

There has also been a series of Alpbach Workshops on Affinity Proteomics that have discussed aspects of antibody generation and characterization. A write up of their 2017 meeting summarized discussions around topics that included antibody specificity being 'context-dependent' and characterization needing to be performed by end users for each specific use (*Taussig et al., 2018*). They also emphasized the fact that characterization data are potentially cell or tissue type specific. They included a summary of efforts underway at the time to establish characterization guidelines, and the roles that publishers and authors can and should play in addressing the optimal use and reporting of antibody-based experiments. At the most recent workshop (March 2024), representatives from various companies presented recombinant antibody or binder generation technologies. Participants

**Table 1.** The 'five pillars' of antibody characterization.

In 2016 an ad hoc International Working Group for Antibody Validation introduced the five pillars of antibody validation/ characterization: (i) genetic strategies; (ii) orthogonal strategies; (iii) (multiple) independent antibody strategies; (iv) recombinant strategies (originally called "expression of tagged proteins"); (v) capture MS strategies (*Uhlen et al., 2016*). In this table each pillar/ strategy (left column) is followed by a brief description of the pillar/strategy, an indication of specificity, example applications for use, and pitfalls. Adapted from *Waldron, 2022* and used with permission.

| | Pillar/strategy | Description | Specificity | Example applications | Pitfalls |
|---|---|---|---|---|---|
| i | Genetic strategies | Knock-out/ knock-down target gene | High | WB, IHC, IF, ELISA, IP | Requires a genetically tractable system and awareness of potential confounders (such as alternative isoforms) |
| ii | Orthogonal strategies | Compare results from Ab-dependent and Ab-independent experiments | Varies | WB, IHC, IF, ELISA | Requires variable expression of the target and cannot entirely rule out non-specific binding to similar proteins |
| iii | Independent antibody strategies | Compare results from experiments using unique Abs to the same target | Medium | WB, IHC, IF, ELISA, IP | Requires the purchase of multiple Abs and knowledge of their epitopes |
| iv | Recombinant strategies | Experimentally increase target protein expression | Medium | WB, IHC, IF | Overexpression of exogenous protein can lead to overconfidence in the specificity of the Ab |
| v | Capture MS strategies | Use MS to identify protein captured by Ab | Low | IP | Requires access to MS and it can be challenging to distinguish between Ab binding target vs protein bound to target |

Ab: antibody; ELISA: enzyme-linked immunosorbent assay; IF: immunofluorescence; IHC: immunohistochemistry; IP: immunoprecipitation; MS: mass spectrometry; WB: Western blotting.

endorsed these, particularly after demonstrations by representatives from YCharOS and Abcam using KO cell lines, which showed that recombinant antibodies were more effective than polyclonal antibodies, and far more reproducible.

An ad hoc International Working Group for Antibody Validation was formed in 2016 with a goal of addressing the "collective need for standards to validate antibody specificity and reproducibility, as well as the need for reporting practices" (*Uhlen et al., 2016*). This group introduced the 'five pillars' of antibody characterization (*Table 1*): (i) genetic strategies (i.e., the use of knockout and knockdown techniques as controls for specificity); (ii) orthogonal strategies (i.e., comparing the results of antibody-dependent and antibody-independent experiments); (iii) multiple (independent) antibody strategies (which compare the results of experiments that use different antibodies to target the same protein); (iv) recombinant strategies (which increase target protein expression); (v) immunocapture MS strategies (in which mass spectrometry is used to identify the protein(s) captured by the antibody). These pillars are not intended to encompass all useful characterization strategies, nor are they all required for each characterization effort: rather, users are encouraged to use as many as feasible (*Uhlen et al., 2016*).

In summary, in order to generate reliable data when using antibodies in an experiment, the characterization of the antibody needs to document the following: (i) that the antibody is binding to the target protein; (ii) that the antibody binds to the target protein when in a complex mixture of proteins (e.g., whole cell lysate or tissue section); (iii) that the antibody does not bind to proteins other than the target protein; (d) that the antibody performs as expected in the experimental conditions used in the specific assay employed. These articles and workshops certainly helped raise awareness of the issues, identified each of the key concerns, and laid out some strategies for improvement.

## History of specific initiatives

Numerous international efforts have been initiated to address challenges in antibody characterization, particularly targeting the human proteome. It is worth noting the incredible value of proteome-wide approaches, not just for studies of human proteins, but for all model organisms used in basic and biomedical research. What is learned from targeting the human proteome will benefit efforts targeting other proteomes, and vice versa.

The early, large-scale efforts typically focused on the use of high-throughput screening and assays, as well as the use of non-antibody binding molecules (such as protein affinity reagents; *Gloriam et al., 2010*; *Taussig et al., 2007*). While these early projects generated some useful reagents and data, they fell short of their initial goals, although they did help to reveal the scale of the challenges and the limitations of the approaches being used. Below is a summary of these efforts plus two related efforts – the Research Resource Identifier (RRID) program, and the Developmental Studies Hybridoma Bank (DSHB) – presented in a somewhat chronological order (*Figure 1*).

### Human Protein Atlas (HPA)

The Human Protein Atlas (HPA), funded in 2003 by the Wallenberg Foundation, is based in Sweden. Its goals include mapping of all human proteins in cells, tissues, organs and blood, using integrated approaches that depend heavily on antibodies. Since its inception it has grown to include other data, including from transcriptomics and RNA-seq (*Human Protein Atlas, 2020*). The first reports generated by the HPA were in 2005 and included data from 718 antibodies generated both by the HPA and those obtained from commercial sources (*Uhlén et al., 2005*; *Nilsson et al., 2005*). Very nice descriptions of different ways antibodies may be characterized, as well as which methods were used in their reports, provided important early data on issues surrounding the goal of generating and characterizing specific antibodies to target the complete human proteome.

Tests for specificity included spotted arrays of 1440 protein fragments (an assay not widely used or available to most researchers) and antigen competition in immunohistochemistry, though each had been found to lack specificity for antibody characterization. Included in this article (*Uhlén et al., 2005*) is the troubling sentence, "For the commercial antibodies, we relied on the quality assurance of antibody providers". In the Discussion the authors point out that they cannot exclude the possibility of cross reactivity to other proteins in their data, and so they encourage ongoing dialog within the scientific community to find the highest quality, validated antibodies and to exclude the bad ones. The HPA still contains much data derived from polyclonal antibodies in which additional targets appear to be recognized. Shortly afterwards they launched antibodypedia. com, a portal for sharing reports on antibodies.

They include data demonstrating that while useful for high-throughput screening of antibodies, signals in peptide or protein displays are poor indicators of success in the more common applications of antibodies (*Björling and Uhlén, 2008*). Thus, early on it was evident that optimal antibody characterization was challenging and lacked consensus from users, but is essential to high quality, reproducible data. This work also noted the key roles to be played by the scientific community as a whole in contributing to characterization efforts and of publicly sharing all such data.

### NeuroMab/NABOR

NeuroMab is a facility at the University of California Davis, with goals that include the generation of mouse monoclonal antibodies – and, more recently, recombinant antibodies – optimized for use in studies of mammalian brains, with emphasis on antibodies useful in immunohistochemistry and Western Blots. Funded by the National Institute of Neurological Disorders and Stroke (which is part of NIH) since 2005, it works with researchers to identify targets relevant to brain studies, and to generate the optimal immunogen(s) and monoclonal antibodies that target them.

The initiative has developed a strategy in which ~1,000 clones or more are screened in two ELISAs in parallel. One ELISA is against the immunogen (typically a purified recombinant protein), and the other is against transfected heterologous cells expressing the antigen of interest that have been fixed and permeabilized using a protocol that mimics that used to prepare brain samples for subsequent evaluation by immunohistochemistry. A large number of positives (typically ~90) move forward for additional testing by immunohistochemistry and Western Blots against brain samples (*Gong et al., 2016*). This greatly increases the chances of obtaining useful reagents as ELISA assays alone may be poor predictors of a reagent useful in other common assays used in research (*Gong et al., 2016*). However, further analyses of this large number of positives are more labor intensive and more costly than the more common practice of analyzing fewer ELISA positive clones, limiting this successful approach to broader application.

NeuroMab also performs a number of other assays, emphasizing immunohistochemistry and Western Blots in rodent brains but also including KO mice, and samples from human brains when possible. This effort is funded and supported by

the neuroscience community in which mouse mutants are commonly used. They also focus on transparency, providing outcomes (both positive and negative) of any evaluation performed, and making the detailed protocols used in evaluation openly available (neuromab.ucdavis.edu/proto-cols.cfm). Thus, the monoclonal antibodies that emerge from the NeuroMab pipeline are essentially already characterized in key assays used by the target researchers, though they emphasize the need to optimize use in each lab and assay employed. They have generated antibodies directed towards more than 800 target proteins, using their system of immunohistochemistry, Western Blots, and immunofluorescence characterization. Later, this work was extended to sequence the VH and VL regions from NeuroMab hybridomas, and to make the sequences publicly available neuromabseq.ucdavis.edu.

NeuroMab has also converted the best antibodies into recombinant antibodies and made the DNA sequences, plasmids for expression, and both monoclonal and recombinant antibodies readily available to researchers through non-profit, open-access sources (*Mitchell et al., 2023*; *Andrews et al., 2019*). The monoclonal antibodies and the hybridomas that produce them are distributed through the DSHB (see below), and the sequences and plasmids for the recombinant antibodies are available at Addgene. It is worth noting here both the value in making sequences of antibodies available to researchers but also the limitations on vendors, as commercial enterprises, who cannot readily do so without risking use of the information by competitors.

More recently, the Neuroscience AntiBody Open Resource (NABOR) was formed. Initially seeded with recombinant antibodies from NeuroMab, it is expanding through partnerships with mission-aligned antibody developers such as the Institute for Protein Innovation. NABOR provides purified recombinant antibodies, the plasmids that encode them, their sequences, and a Data Hub for detailed data and protocols, through Addgene. NABOR antibodies are distinctive in that they not only have a RRID number assigned to them, but their sequences are also openly available. While RRIDs go a long way towards improving reproducibility, they fall short in that they do not rely on sequence data. Historically, antibody manufacturers rebrand or out-license clones to distributors, and over time antibody names may change. This makes it possible to have different RRIDs for the same antibody, and is particularly problematic as antibodies are

frequently cross-validated by comparing staining patterns of 'unique' clones. Furthermore, RRIDs do not take into account different lot numbers, which is particularly important when dealing with polyclonal antibodies. A scientist may think they have validated their antibody with a different clone when, in reality, they have compared it to itself. Only with open sequences will a scientist know the precise molecular identity of the tools they are using. However, given the reluctance of antibody companies to release antibody sequences, or antigen(s) used to generate antibodies, specific identifier codes akin to RRIDs could be linked to specific sequences, so at least scientists would know if they were using antibodies with the same sequence.

If one were starting the task of generating and validating antibodies targeting the entire human proteome, the NeuroMab/NABOR pipeline represents a great model, although one that may be difficult to scale to the entire proteome. Yet the generation, characterization, and distribution of new recombinant antibodies does not address the concerns arising from the large numbers of antibodies being sold and used today that lack appropriate characterization and can lead to the publication of non-reproducible science.

The next two large-scale projects, described below – the Protein Capture Reagent Program and Affinomics – were broader in aspirations and, perhaps as a result, were less sustainable, and did not result in as many publications as might be expected, but they were still important and instructive exercises.

### Protein Capture Reagent Program (PCRP)
Concerted efforts toward the goals of a collection of monoclonal antibodies covering the human proteome ramped up dramatically in 2010 when the NIH funded the Protein Capture Reagent Program (PCRP) for five years. This program focused on the generation and characterization of monoclonal antibodies and recombinant antibodies targeting human transcription factors (*Roy et al., 2021*; *Venkataraman et al., 2018*; *Lai et al., 2021*; *Blackshaw et al., 2016*). The focus on transcription factors was viewed as a test case that, if successful, would be scaled toward the entire proteome in subsequent years.

The PCRP involved bacterial expression of antigens, used to inoculate animals or screen recombinants, identification of high affinity reagents, and characterization assays. The choice of transcription factors as sole targets made extrapolation to the rest of the human proteome less

generalizable. Note that most of the effort and costs incurred went into generating antibodies rather than characterizing existing ones. Among specific outputs from this program are a collection of 1406 monoclonal antibodies (available through the DSHB) targeting 737 human proteins (*Venkataraman et al., 2018*; *Lai et al., 2021*). The PCRP was also responsible for a number of spin-off initiatives, such as the Recombinant Antibody Network.

Although the PCRP did not launch the concerted effort to generate and validate affinity reagents to target the entire human proteome, it brought to light the magnitude of the goals – that is, the challenges involved in (i) generating high quality antigens; (ii) generating appropriate recombinant antibodies; (iii) identifying the high affinity and highly specific reagents required; (iv) characterizing those antibodies at least in the most common assays that they will be used in; (v) making all data readily available to the public.

### Affinomics

The EU-funded Affinomics program grew out of two earlier projects: ProteomeBinders (*Taussig et al., 2007*) and AffinityProteome (*Stoevesandt and Taussig, 2012a*). Like the PCRP, it shared the ultimate goal of generating, screening, and validating a collection of protein binding reagents useful for characterization and analyses of each member of the human proteome (*Stoevesandt and Taussig, 2012b*). It identified a group of target proteins for initial testing that included protein kinases, SH2 domain containing proteins, protein tyrosine kinases, proteins known to be mutated in cancer, and cancer biomarkers (*Landegren, 2016*; *Stoevesandt and Taussig, 2012b*). The goals of Affinomics were broken down into seven areas: (i) protein/antigen production; (ii) binder production (not limited to antibodies); (iii) binder characterization (microarrays, Western Blots, and immunofluorescence); (iv) optimization of affinity reagent selection technologies; (v) development of tools for analysis of human serum for cancer markers; (vi) interactomes of a few, key oncogenes (*e.g.*, RAS, mTOR, EGFR); (vii) development of suitable databases for reporting results. The 286 antibodies generated have been shared and are available through the DSHB.

Together, the PCRP and Affinomics programs have helped to highlight the need for good, validated antibodies, and the challenges associated with high-throughput approaches, particularly those that use assays that are not widely used in research labs.

### CiteAb

CiteAb is an online database that allows researchers to search for antibodies and other reagents that are currently available. It began as a data mining research project in the laboratory of Andrew Chalmers at the University of Bath in 2012. Through data mining coupled with manual reviewing of the literature and collaborations with vendors of antibodies, it quickly grew and today covers over 14 million reagents with links to 6 million citations. Collecting such data can clearly help end users identify and begin to evaluate potential reagents for their experiments, but users should also be aware of the limitations of depending upon citation numbers as evidence of value in the absence of actual characterization data (see, for example, *Laflamme et al., 2019*; *CiteAb, 2024*). Despite this, using provided filters to restrict searches to user needs can allow one to generate a short and manageable list of potential reagents to consider using.

In 2019 CiteAb began linking reagents to published images (e.g., immunofluorescence and Western blot data) to provide end users with more information, and it recently added links to characterization data generated by YCharOS to antibody pages (*Longworth, 2024*). CiteAb has also organized three International Antibody Validation meetings.

### Research Resource Identifier (RRID)

The Research Resource Identifier (RRID) program can generate unique identifiers for antibodies and other reagents (*Menke et al., 2020*; *Bandrowski et al., 2016*; *Bandrowski et al., 2023*). Users can also search the RRID website to find, cite, and deposit characterization data, although the RRID initiative does not perform any antibody characterization studies. It is also important to note that an antibody can be sold by multiple different vendors with each using a different RRID number, and different lots of the same manufacturer's antibody will have the same RRID, even if there may be significant lot-to-lot variation. To the extent possible, this practice should be replaced by better practices, including assigning one RRID to a reagent so that each antibody is given one and only one RRID number.

Data mining of published articles was used early on to document problems in the underreporting of antibodies, cell lines and model organisms (*Vasilevsky et al., 2013*). The neuroscience community played a large role, via the Neuroscience Information Framework (NIF) and the Antibody Registry (*Bandrowski et al., 2023*).

With the support of a number of journals and NIH funding, the RRID has seen steady increases in use and its impact on data reproducibility. By 2017 >5,000 articles in >380 journals were including RRID data and today those numbers are much higher. Notably, an initiative similar to RRID, termed the antibody identity card (Ab ID Card) was also begun, though it appears to have gained less traction than RRID. Another important development from these efforts is SciScore (*Menke et al., 2020*), an algorithm that can quickly search through text to identify the presence, or lack, of important identifying information for the reagents used. Use of such a tool by authors, journals and reviewers would facilitate and speed the inclusion of key information/identifiers, improve the reproducibility of any work that uses it, and decrease the burden on end users.

### Developmental Studies Hybridoma Bank (DSHB)

Since its inception in 1986, the Developmental Studies Hybridoma Bank (DSHB) at the University of Iowa has maintained and distributed at minimal cost hybridoma cell lines and monoclonal antibodies shared by investigators. DSHB ensures the availability of antibodies for both human and non-human model organisms, which is particularly important to smaller fields of research. Many important antibodies that are widely used in studies of human muscle, cancers and neuroscience, and in key model organisms are found exclusively at the DSHB. Over 6,000 hybridoma lines – including those submitted by the PCRP, Affinomics, CPTAC, and NeuroMab initiatives – are maintained by DSHB, which also manages more than 600 recombinant antibodies, many of which come from the CPTAC.

With >65,000 samples distributed in the past year, the DSHB remains an important component in the availability of research antibodies. However, like many commercial vendors, it cannot do the characterization work on the reagents that they distribute. In some respects, the antibodies in DSHB's collection reflect the characterization crisis. Many of the antibodies donated to DSHB during the initial wave of hybridoma generation (<~2005) were made in research labs and are extensively characterized. In contrast, the efforts of many of the later, high-throughput projects remain less well characterized. DSHB was originally funded by the National Institute of Child Health and Human Development (which is part of NIH), but it has been fully supported by user fees since 1997.

### Clinical Proteomic Tumor Analysis Consortium (CPTAC)

Biomarkers are needed to speed the diagnosis and treatment of diseases, notably cancers. Antibodies are used to identify and validate biomarkers, and to create assays for the detection of biomarkers and, potentially, the therapeutic targeting of biomarkers. The National Cancer Institute (part of the NIH) set up the Clinical Proteomic Tumor Analysis Consortium (CPTAC) in 2006 to fund both intramural and extramural projects. One notable intramural project is the Antibody Characterization Laboratory (ACL), which makes and characterizes renewable antibodies for use in cancer-related research using a combination of ELISA, Western Blots, immunohistochemistry and other assays, although it does not currently use KO cell lines to characterize reagents. At the time of writing, the ACL has developed 946 antibodies targeting 570 antigens (proteomics.cancer.gov/antibody-portal), which are obtainable from the DSHB.

### Antibody Characterization through Open Science (YCharOS)

The Structural Genomics Consortium (SGC) began in 2003 with a focus on the determination of protein structures for the human proteome, and after considerable success in that arena, moved into other areas, including antibody generation and characterization. In 2020 the Antibody Characterization through Open Science, or YCharOS initiative, was launched at the Montreal Neurological Institute at McGill University as part of the SGC, with a focus on the characterization of existing antibodies. There is also a Canadian company called YCharOS Inc that raises funds for antibody characterization studies.

YCharOS has refined an approach (*Laflamme et al., 2019*) based upon the use of KO cell lines to test antibodies in Western Blots, immunoprecipitation and immunofluorescence. As a result of ongoing collaborations with 12 industry partners and additional academic researchers, it has developed consensus protocols for each of these techniques, and these protocols can be used widely in antibody characterization efforts (*Ayoubi et al., 2024*). As of March 2023, YCharOS has reported results from the testing of more than 1,000 antibodies and had published 96 antibody characterization reports (one report per protein) at zenodo.org/communities/ycharos, and a number of peer-reviewed articles at f1000research.com/ycharos.

The YCharOS group recently published a highly impactful study that analyzed a set of 614 antibodies targeting 65 proteins (*Ayoubi et al., 2023*). For example, the group found that 50–75% of the protein set was covered by at least one high-performing commercial antibody, depending on the application. Extrapolation to the human proteome suggests that commercial catalogs contain specific and renewable antibodies for more than half of the proteome, confirming an initial YCharOS hypothesis. This study also showed the use of KO cell lines to be superior to other types of controls for Western Blots, and even more so for immunofluorescence imaging. Shockingly, it also revealed that an average of ~12 publications per protein target included data from an antibody that failed to recognize the relevant target protein!

In addition to highlighting the magnitude of the antibody crisis, this work also revealed the value and importance of industry/researcher partnerships, as both the antibodies and the KO cell lines were donated by vendors. These vendors also evaluated the resulting data, and often re-evaluated their own in-house data. These vendors proactively removed ~20% of the antibodies tested that failed to meet expectations, and modified the proposed applications for ~40% (*Ayoubi et al., 2023*). Thus, these data demonstrate the means for both identifying useful reagents and removing bad ones. A key challenge is scaling up such efforts to proteome scale. Finally, this study showed the value of recombinant antibodies, demonstrating that on average they outperformed both monoclonal and polyclonal antibodies in all the assays used. Note that the failure of an antibody to work in any assay (or in a small number of assays) does not mean that it should necessarily be removed from the market as it may work in other assays. However, the burden should be on the vendor to make those data known to potential end users.

### Only Good Antibodies (OGA)

Set up in 2023, and based at the University of Leicester, the Only Good Antibodies (OGA; *Biddle et al., 2024*) community works with and helps to promote the YCharOS pipeline. OGA started as a partnership between biomedical researchers and behavioral scientists who used antibodies in their research, and has the following aims: (i) to promote awareness of the issues surrounding the use of antibodies in research; (ii) to help educate researchers; (iii) to ensure better availability of characterization data; (iv) to aid the planning for antibody characterization as part of research funding proposals; (v) to better share data with reporting in publications and open data repositories. OGA has organized and run educational workshops and webinars as part of its awareness campaign, and it recently co-organized (with NC3R) a workshop in London titled 'Defining the role of antibodies in improving research reproducibility', that was attended by a range of stakeholders (including many of the authors of this article).

## Technology development

Before embarking on large, publicly-funded projects, a crucial question is whether to wait for new technologies that might expedite success, enhance output quality, or reduce costs. This dilemma was evident in the human genome sequencing discussions, where starting the project proved valuable despite initial technological limitations. Note that there is a clear difference between the work involved in *generating* optimal antibody reagents and in *characterizing* them. This prompts the question: Is there a need to wait for newer technologies that will expedite and improve the processes involved in antibody characterization?

The primary challenge in characterizing antibodies lies in the diversity of assays for antibody usage, the variations within each assay, and the incompatibility of common assays with high-throughput methods. To establish realistic goals, one needs a finite set of assays, with standardized protocols agreed upon by stakeholders. These characterization efforts should be performed using assays similar to those used by end users.

Towards this end, members of the YCharOS team and representatives from ten leading antibody manufacturers recently co-wrote a method article that contains detailed protocols for Western Blots, immunoprecipitation and immunofluorescence (*Ayoubi et al., 2024*). A consensus on such assays – made possible by collaborations among stakeholders – could lead to a pipeline identifying reagents that are effective for these assays. Most antibodies will work in some assays but not all. Thus, even if an antibody does not perform well in some characterization trials, it may be valuable in other assays or under other conditions than those used previously. Thus, all data used in these evaluations, and the protocols used, should be openly shared to allow end users to assess the data as they deem most appropriate to their needs. Even with such

a pipeline in place, end users must still validate the antibodies they plan to use in their own work.

A challenge in large-scale antibody characterization is the current lack of high-throughput assays (including ones that might be readily automated) that are similar enough to end user assays to be optimal for characterization. While peptide or fragment display approaches exist, extrapolating their data to more common assays is risky. However, a combination of immunoprecipitation and mass spectrometry can be used to speed and improve the characterization of the selectivity and specificity of antibodies at scale in this one use (*Marcon et al., 2015*).

In contrast to the limited role that high-throughput assays are likely to play in the characterization of antibodies, technologies allowing rapid screening of large libraries for high-affinity, specific recombinant antibodies are generating large numbers of new antibodies, some with affinities better than those obtainable by immunization (*Azevedo Reis Teixeira et al., 2021*). These recombinant antibodies will still require characterization in all intended assays, but once characterized will significantly ameliorate problems related to lot to lot variation. We can expect the number of commercially available recombinant antibodies to continue to grow rapidly in the coming years, only further arguing for the need to gain consensus on issues surrounding their characterization, proper use, and reporting.

Finally, a role for technology in this field that will likely play increasingly important roles in the future development and optimization of antibodies will come from deep learning models, such as AlphaFold. These models should enable predictions of the antibody-antigen complex, aid in the identification of the epitope targeted, and help determine if folding, post-translational modifications or other issues may influence the output from use of the antibody. These future prospects are another reason to encourage open access to the antibody sequences of recombinant antibodies.

## Stakeholder roles and ways forward

For many years end users have been angered by the waste of time and money associated with bad antibodies, and have been frustrated by the inability to reproduce work from other labs.

The goals of providing researchers with optimal reagents and enhancing the reproducibility of antibody-based data depends on all the relevant stakeholders understanding the issues involved and working collaboratively towards improvement. At a minimum this involves: (i) development of consensus in characterization standards; (ii) the use of open source(s) for depositing characterization data, thereby ensuring access and reducing redundancy; (iii) rigorous use of reporting tools to clearly and unambiguously identify reagents used (such as RRID and SciScore); (iv) prioritized engagement with active participants to incentivize and speed the adoption of best practices for reproducibility; and (v) the widespread adoption of recombinant antibodies.

We list below the various stakeholders, along with recommendations for what they can do to improve the current situation. In an ideal long-term scenario, only recombinant antibodies should be used, and they should be characterized for specificity by testing on KO cell lines in the specific assays in which they will used. While it would be ideal to have antibody sequences publicly available, this remains unlikely for commercially sourced ones. That said, it appears useful to explore concepts around providing RRID-like identifiers to recombinant antibodies known to have identical sequences, allowing researchers to at least know they are using antibodies of identical sequence, without knowing the sequences themselves.

### Researchers and end users

Researchers are responsible for performing adequate controls to ensure reagent performance, enabling confident and accurate interpretations and conclusions, in both academic and biotech/pharma environments. This means that scientific papers must include detailed methods sections, as well as unambiguous descriptions of the antibodies used (RRID, source, catalog number, details as to the type of antibody, immunogen used to raise the antibody, protein concentrations used in each assay) and means of characterization. Where possible, end users should avoid the use of polyclonals, and use well characterized recombinant antibodies instead. End users can also help improve vendor websites by providing feedback, both good and bad, including data from the use of their products. Due to the large number of variations in assays between labs, there can be no single gold standard for use of an antibody in any assay. Rather, optimal controls should be included as a routine part of the data presentation, to support sound interpretations.

While selecting antibodies, researchers should rely on online databases (which are improving),

rather than relying solely on vendor information, which can often be incomplete or misleading. We list here four guides to help users identify the best antibody reagents.

- Roncador et al. walk the reader through steps suggested by the European Monoclonal Antibody Network (https://www.euromabnet.com/) to find optimal reagents (*Roncador et al., 2016*).
- Acharya et al. describe recommended steps for selecting, validating, and reporting on the use of antibodies (*Acharya et al., 2017*).
- The Antibody Society has recorded a series of ten webinars that discuss the issues involved (*Voskuil et al., 2020*).
- The CiteAb website contains a good summary of the issues surrounding the purchase and use of antibodies (*CiteAb, 2024*).

Many antibodies are 're-sold' to multiple vendors, each of which will use their own catalog number or characterization data (often copied from others). Avoiding such re-sellers should be a priority, and buying from the original source, when it can be identified, can improve both the quality of the data available and the responsiveness of the vendor to queries.

Researchers can also, when reviewing papers and grants, insist that authors and applicants adequately characterize their reagents, and also report how they did this in full. Likewise, the experimental methods used in a study should be "described in sufficient detail to allow another researcher to reproduce the work" (*Malički, 2024*). Editors, reviewers and authors should all share this goal.

As experts in their fields, researchers are ideally suited to work with others in the same field to generate and extend the basic characterization data from open sources into assays that could become important to that particular field. These assays will vary with the class of protein, but good assays will help a field by improving the quality of data and helping to identify the best – and worst – antibodies. Funding agencies appear unwilling to support new, large-scale antibody characterization projects, but perhaps they will support more focused projects that involve the experts in a field first prioritizing the key proteins in that field, generating or collecting appropriate KO cell lines, working together to characterize available antibodies, and then sharing the results. Discussing such efforts at scientific meetings would be time well spent. Finally, anyone writing a grant application in a field that lacks adequate antibodies needed for key experiments, should consider including requests for funding to generate and characterize such antibodies, explicitly making both the data and antibodies available to others.

## Universities

Institutions should ensure that students, post-docs and staff all receive comprehensive training in the use of reagents, including antibodies. This includes both the technical aspects and the interpretation of experimental results, along with optimal controls. Existing resources, like the Antibody Society's webinar series (*Voskuil et al., 2020*), can support curriculum development in this area. Universities can also work with non-profits like YCharOS to promote scaling up their efforts. Often universities contain concentrations of expertise in different areas of research or protein families that could be leveraged to obtain funding for characterization work, ideally using comparable protocols to what is being done at YCharOS.

## Journals/Publishers

Journals play a crucial role in establishing and maintaining high research standards, so it is not clear why they have been so slow to adopt standards for reporting the use of antibodies and for ensuring that appropriate controls were performed. The Journal of Comparative Neurology was among the first to clearly describe both the need for antibody information in manuscripts and the details of how to include it in a methods section (*Saper, 2005*). A unified approach to describing the use of antibodies in manuscripts, including RRID numbers and protocol details, should be a required part of the submission and review process at all reputable journals. Authors should also be required to report the amount of antibody used in each assay in protein concentrations (rather than dilution, which is ambiguous). We also encourage journals to use algorithms (such as SciScore) to automate this process, and therefore lower the burden on authors, reviewers, and editors.

Editors and publishers may be resistant to develop and enforce reporting rules for antibodies, as authors may interpret them as extra work that discourages submissions at such a journal. We encourage journals to establish and enforce the highest standards of reproducibility in the work they publish, and authors to submit their work to those journals that demonstrate such high standards.

### Antibody vendors and repositories

It is important to acknowledge that commercial vendors of antibodies are businesses and, as such, they are motivated by profits. Vendors, as well as the DSHB and Addgene, should accurately represent their products, including comprehensive information for users to evaluate antibodies before purchase whenever available, including sufficient details of any data shown to allow accurate interpretation. Collaborative efforts with groups like YCharOS to validate and openly report antibody performance are vital to obtain the characterization data required. Vendors should update their data regularly and remove ineffective antibodies from the market (and, where possible, end users should buy their antibodies from such companies). Vendors should also take the lead in ensuring that each antibody is assigned one, and only one, RRID to allow better tracking and linkage to characterization data. When distributing to other vendors, they should have the opportunity to make this a requirement.

### Societies

Scientific societies play important roles in education, training, advocacy, and other activities through meetings, workshops, newsletters, and journals. The most recent annual meeting American Society for Cell Biology (ASCB) included a workshop on Antibody Validation that was attended by representatives from all stakeholders, and we encourage the inclusion of such activities in such meetings to further raise the awareness of the issues and to encourage training in best practices. Societies can also organize expert groups to discuss how best to characterize specific types of antibodies.

### Disease foundations

A number of disease foundations have recognized the importance of working with researchers to help identify and make available antibodies that target key proteins and pathways implicated in a particular disease pathology. A better understanding of the ways in which changes in protein function or location may contribute to the pathology will help researchers working on treatments for the disease. For example, The Michael J Fox Foundation for Parkinson's Research (MJFF) has developed a unique Research Tools Program which focuses solely on the generation, characterization/validation, and open distribution of preclinical laboratory tools and models for Parkinson's disease research. Partnering with both manufacturers/vendors and academic researchers who are experts in particular targets, MJFF has made available 200 research tools to date (see, for example, *Davies et al., 2013*). MJFF also funds the characterization of commercial reagents through groups like YCharOS and academic labs to ensure information is readily available. We encourage more foundations to consider programs such as this and, more generally, to think about how they might support ongoing efforts to improve the rigor and reproducibility of research relevant to their work.

### Funding agencies

While federal funding agencies argue they are not regulatory bodies, they play a crucial role in supporting high-quality, reproducible research. Requirements for data sharing, conflict-of-interest reporting, training in ethics, and animal care, introduced by funders have made research stronger. We encourage funding agencies to develop new opportunities to support efforts focused on antibody characterization, including training in the use of key reagents. Their support for a repository for KO cell lines is just one example of straightforward ways they could support efforts to improve the quality and reproducibility of research. The development and support for consortia of stakeholders is viewed as the best way to move forward, though ongoing support for individuals performing antibody generation, characterization, and distribution is also strongly encouraged. Building upon and expanding such efforts should be a high priority.

## Summary

Over the past two decades, many individuals and projects have contributed significantly to addressing the crisis stemming from inadequate characterization of antibodies used in research. Efforts to generate and make available antibodies for the entire human proteome have illuminated the complexities of the issues, emphasizing the need for a multi-faceted approach and the involvement of all stakeholders. Key achievements include bringing antibody characterization to the forefront of stakeholder concerns, and separating the objectives of antibody generation from characterization. Newer technologies promise to accelerate the creation of new recombinant antibodies (so researchers would no longer have to use polyclonal antibodies). However, the characterization of these new reagents – and the six million or so antibodies currently on the market – remains crucial, necessitating data

sharing, identification of optimal reagents, and the removal of non-specific antibodies. Let us be clear, there is no one 'fix' to the problems outlined here. The problems are ongoing and will continue. Raising awareness of the issues and holding all stakeholders accountable for contributing to improvements are viewed as the best approaches to minimizing future waste and damage.

Each stakeholder has a distinct motivation yet a shared commitment to supporting well-controlled antibody-based studies. Researchers strive for high-quality data to drive novel discoveries, quality publications, and paths towards improving human health. Universities aim to support research and provide comprehensive training. Disease foundations focus on identifying key reagents for disease pathways, while reputable journals and publishers aim to uphold high scientific standards. Vendors have the responsibility to provide characterized products, a goal some are now pursuing through collaborations with groups like YCharOS and use of RRIDs. Funding agencies have supported initiatives in the past and should continue to promote high-quality, reproducible research, taking advantage of what has been learned from prior projects and building upon it to improve the quality of the reagents and the resulting data.

The community is encouraged to support and further develop pipelines like the YCharOS pipeline, which exemplify collaborative efforts towards consensus on characterization assays and public data sharing. The use of tools like RRID and SciScore should become standard practice, easing the burden of enforcing necessary changes and enhancing the overall quality and reproducibility of antibody-based research. Scientists can build consortia in their areas of expertise to share the costs of characterization and make readily available the best possible reagents with agreed upon characterization testing and uses.

## Acknowledgements
The authors would like to thank the many scientists who have been involved in working to improve the quality and reproducibility of research that relies on antibodies. Many of the authors attended a meeting at the Wellcome Trust in London in February 2024 ('Defining the role of antibodies in improving research reproducibility') that was supported by The University of Leicester BBSRC Impact Accelerator Account, the Institute for Protein Innovation, the NC3R, and the Structural Genomics Consortium, and the discussions at that meeting have contributed to this article.

**Richard A Kahn** is in the Department of Biochemistry, Emory University School of Medicine, Atlanta, United States
rkahn@emory.edu
https://orcid.org/0000-0002-0259-0601

**Harvinder Virk** is in the Department of Respiratory Sciences, University of Leicester, Leicester, United Kingdom
https://orcid.org/0000-0002-9739-9593

**Carl Laflamme** is in the Department of Neurology and Neurosurgery, Structural Genomics Consortium, The Montreal Neurological Institute, McGill University, Montreal, Canada
https://orcid.org/0000-0001-5906-025X

**Douglas W Houston** is at the Development Studies Hybridoma Databank, University of Iowa, Iowa City, United States
https://orcid.org/0000-0002-7438-4655

**Nicole K Polinski** is at The Michael J Fox Foundation for Parkinson's Research, New York, United States
https://orcid.org/0000-0003-4613-9115

**Rob Meijers** is at the Institute for Protein Innovation, Boston, United States

**Allan I Levey** is in the Department of Neurology, Emory University School of Medicine, Atlanta, United States
https://orcid.org/0000-0002-3153-502X

**Clifford B Saper** is in the Department of Neurology and Program in Neuroscience, Harvard Medical School and Beth Israel Deaconess Medical Center, Boston, United States

**Timothy M Errington** is at the Center for Open Science, Charlottesville, United States
https://orcid.org/0000-0002-4959-5143

**Rachel E Turn** is in the Department of Microbiology and Immunology, Stanford University School of Medicine, Stanford, United States
https://orcid.org/0000-0001-5389-4560

**Anita Bandrowski** is in the Department of Neuroscience, University of California at San Diego, La Jolla, United States
https://orcid.org/0000-0002-5497-0243

**James S Trimmer** is in the Department of Physiology and Membrane Biology, University of California Davis School of Medicine, Davis, United States
https://orcid.org/0000-0002-6117-3912

**Meghan Rego** is at Addgene, Watertown, United States

**Leonard P Freedman** is in the Frederick National Laboratory for Cancer Research, Frederick, United States
https://orcid.org/0009-0007-5726-150X

**Fortunato Ferrara** is at Specifica, a $Q^2$ company, Santa Fe, United States

**Andrew RM Bradbury** is at Specifica, a $Q^2$ company, Santa Fe, United States

**Hannah Cable** is in the Department of Research and Development, Abcam, Cambridge, United Kingdom

https://orcid.org/0009-0001-0971-5761

**Skye Longworth** is at CiteAb, Bath, United Kingdom

*Author contributions:* Richard A Kahn, Conceptualization, Writing – original draft, Project administration, Writing – review and editing; Harvinder Virk, Conceptualization, Writing – original draft, Writing – review and editing; Carl Laflamme, Conceptualization, Writing – original draft, Writing – review and editing, Drafted the section on YCharOS; Douglas W Houston, Writing – original draft, Writing – review and editing; Nicole K Polinski, Writing – original draft, Writing – review and editing, Drafted the section on the Michael J Fox Foundation; Rob Meijers, Writing – original draft, Writing – review and editing; Allan I Levey, Writing – review and editing; Clifford B Saper, Writing – review and editing; Timothy M Errington, Writing – review and editing; Rachel E Turn, Writing – review and editing; Anita Bandrowski, Writing – review and editing; James S Trimmer, Writing – review and editing, Drafted the section on NeuroMab/NABOR; Meghan Rego, Writing – review and editing, Drafted the section on Addgene; Leonard P Freedman, Writing – review and editing; Fortunato Ferrara, Visualization, Writing – review and editing; Andrew RM Bradbury, Writing – review and editing; Hannah Cable, Writing – review and editing; Skye Longworth, Writing – review and editing

*Competing interests:* Anita Bandrowski: Co-founder and serving as CEO of SciCrunch Inc, a company that works with publishers to improve the scientific literature; the terms of this agreement have been approved by the COI office at the University of California at San Diego. Meghan Rego: Employed by Addgene, a company that may be affected financially by the opinions reported in this article. Hannah Cable: Employed by Abcam, a for-profit company that sells antibodies and which may be affected financially by the opinions reported in this article. The other authors declare that no competing interests exist.

## Funding

| Funder | Grant reference number | Author |
|---|---|---|
| National Institute of General Medical Sciences | 1R35GM122568 | Richard A Kahn |
| Genome Canada, Genome Quebec and Ontario Genomics | OGI-210 | Carl Laflamme |
| National Institutes of Health | U54AG065187 | Allan I Levey |
| National Institutes of Health | U24NS109113 | James S Trimmer |
| National Institutes of Health | U24NS119916 | James S Trimmer Meghan Rego |

The funders had no role in study design, data collection and interpretation, or the decision to submit the work for publication.

## Decision letter and Author response

Decision letter https://doi.org/10.7554/eLife.100211.sa1
Author response https://doi.org/10.7554/eLife.100211.sa2

## Data availability

No data were generated for this work.

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
