## [Decision Letter]

**Decision letter after peer review:**

Thank you for submitting your article ‘Antibody Characterization: A Historical Overview of Approaches to Enhancing Reproducibility in Biomedical Research and Ways Forward’ to *eLife* for consideration as a Feature Article. Your article has been reviewed by two peer reviewers, and the evaluation has been overseen by the Chief Magazine Editor of *eLife* (Dr Peter Rodgers). The following individuals involved in review of your submission have agreed to reveal their identity: Fridtjof Lund-Johansen and Dario Alessi.

The reviewers and editor have drafted this decision letter to help you prepare a revised submission.

*Reviewer #1:*

Summary

The authors have written an historical background for the problems with the quality of research antibodies and describe initiatives taken to improve the situation. There has indeed been a very positive development during the past years, and a discussion of how this has happened is important. The focus is on the authors' own initiatives. This is natural, and I think many readers are unaware of wonderful resources such as the DSHB, Neuromab and NABOR and YCharOS.

Personally, I would prefer a more in-depth analysis of the current problems and more specific suggestions for how to solve them.

[1.1] Are there really 6 million antibodies on the market. How many have ever been sold or used in publications? As for the estimated financial loss of up to $1.8bn in the US per year: how confident are the authors of this estimate? People in the business have told me that most of their revenue comes from sales of a few top-selling products: it would be good if some of the authors who are based in industry could comment on this.

[1.2] The authors suggest that it is crucial to test all 6 million antibodies in multiple applications. I doubt that anyone thinks this is realistic or necessary. I would encourage them to come up with ideas for a more realistic strategy. There has to be a pre-selection step. What will it be and who will select? The next questions are who will test and how should the results be made available? My personal opinion is that money should be spent on making KO cells for as many genes as possible starting with the most widely studied proteins and make the cells available to all.

[1.3] Until now, the industry standard for primary screening has largely been western blotting. The implication is that we have discarded antibodies that bind to conformational epitopes. With KO cells, we can look beyond WB. How will that impact antibody development? The authors explain that there is a clear difference between the work involved in generating optimal antibody reagents and in characterizing them. I would argue that the two are closely linked and that you get what you screen for. I therefore find it interesting that Neuromab uses ELISA for the primary screen. I am sure that a lot of readers would be interested in learning more about their pipeline.

[1.4] The topic of webportals is important and deserves attention. A successful portal can have immense market power. The authors mention the Antibodypedia (from the Human Protein Atlas), the Antibody Registry, the Euromabnet and YCharOS. The initiatives are great, but to me it looks as if CiteAb has already established a position as the analogue of Clarivate Analytics for antibodies. The authors may disagree, and the landscape can change quickly in the era of ChatGPT. Still, I don't think it is wise to ignore CiteAb's role in the current market.

*Reviewer #2:*

Summary

This is an important review that comprehensively discusses approaches to enhancing the reproducibility of antibody characterization. The authors are the key researchers working in this area from researchers to funding bodies and foundations to antibody vendors.

The authors highlight the huge amounts of resources that are lost each year and the resulting poor science that is undertaken leading to unreproducible results from using antibodies that are poorly characterised and have not been sufficiently validated. The study provides a very good historical context, discussing many previous past efforts and more recent efforts to improve the rigour with which antibodies are characterised. These include the Human Protein Atlas, the Neuromab Facility, The Protein Capture Reagent Programme (PCRP), Affinomics, an EU-funded programme. The authors also discuss in depth the ‘Research Resource Identifier’ (RRID) Programme that provides a unique identifier to each antibody.

The authors also highlight the Developmental Studies Hybridoma Bank (DSHB) that has played such an important role in distributing hybridoma clones and more recently recombinant antibody cDNA vectors. The authors also discussed the Clinical Proteome Tumour Analysis Consortium (CPTAC) that provides antibodies for cancer research. The authors also discuss a very valuable effort, started by the Structural Genome Consortium, called the Antibody Characterisation Through Open Science (YCharOS). This consortium have rigorously tested over 1000 antibodies and characterised detailed antibody reports on many of these, and have found that a significant proportion of all antibodies that have been used in many previous publications shockingly do not detect endogenous proteins.

The authors also discuss a community called Only Good Antibodies (OGA) that are also attempting to validate antibodies in a consistent manner and provide this information to the research community. The authors also highlight the important roles that researchers, vendors, universities, journals, funding society and disease foundations can have to mandate that researchers only use well characterised antibodies, or if the antibody hasn't been well characterised previously, the authors need to undertake essential characterisation, demonstrating that the antibodies they are using are indeed recognising the designated target, preferably using a knockout or knockdown approach. The authors also highlight important work undertaken by the Michael J. Fox Foundation for Parkinson's research where they've developed a unique “research tools programme” which focuses on characterisation and validation of antibodies used for research. Thus far, 200 research tools have been produced to date that are all highly characterised, and groups using these reagents can be reassured that the antibodies are very well characterised.

The authors also discuss the importance of trying to move away from polyclonal antibodies to recombinant antibodies in which the sequences are known and antibodies can be expressed more consistently in human cell lines. Ideally having sequences available for antibodies certainly helps researchers to ensure that the antibody is the correct reagent and enables them to express these reagents as scale in their laboratories, rather than having to purchase these critical reagemts at high cost from a vendor.

The authors provide a lot of important suggestions and recommendations including improving awareness and training in laboratories of the importance of rigorous antibody characterization has in ensuring the reproducibility of life sciences research.

This article is a must read for Life Scientists at any stages of their careers who make use of antibodies for their research. I recommend acceptance of this study.

[2.1] The authors highlight the major problem with companies that distribute antibodies to multiple other companies resulting in the same antibodies having different RRID numbers, that causes major confusion. It would be important to put in place a system whereby if a company purchases an antibody from another vendor that they tried to ensure that the original RRID number is used for this purpose-this may be difficult to achieve. The authors should comment on this.

Linking antibodies with their sequences would also be good but also unlikely to work as most antibody companies do not disclose the sequences of their antibodies for commercial reasons. It would be good if the authors could also comment on this.

[2.2] YCharOS found that on average 12 publications per protein target contain data from an antibody that failed to recognise the endogenous targets. YCharOS is an extremely important effort and I hope that it will be able to continue expanding their efforts to properly validate many of the widely used antibodies. This effort represents a drop in the ocean and the feasibility of dramatically scalling up the efforts of YCharOS to test a greater proportion of all widely used antibodies seems very challenging.

It would be good if the authors could say more about the challenges involved in scaling up such efforts.

[2.3] It would be helpful to have a table or figure outlining the key steps/criteria that the authors believe are needed to thoroughly characterise antibodies for immunoblotting, immunofluorescence, immunoprecipitation, immunohistochemistry, and flow cytometry. This might help students and postdocs plan their experiments to ensure that the data required to validate the antibodies that they're using is available. Ideally, vendors would also sign up to this level of characterisation data being provided in the antibody datasheets. Ideally third-party companies or consortia could be available to undertake this important QC work in the future and feed the back to the community.

[2.4] If the sequences of the recombinant antibody are known, using Alphafold 3 it should be possible to predict the structure of the antibody antigen complex. This might enable researchers to generate subtle point mutations in the recombinant antibody that ablates their affinity for the intended target. Such mutated antibodies could serve as useful control reagents for certain applications such as IF, IHC, flow cytometry, IP, to ensure that the signals that are observed are indeed mediated by binding to the antigen and not to a non-specific site.

It would be good if the authors could comment on this (including the possibility that vendors might start providing these control reagents in the future).

[2.5] An important recommendation that is not made in the review (perhaps because vendors are co-authors) is that vendors should sign up to withdraw all antibodies found not to work or detect the endogenous target by trusted third party testers or consortia. Again it would be good to discuss this issue in the review.

---

## [Author Response]

Reviewer #1:SummaryThe authors have written an historical background for the problems with the quality of research antibodies and describe initiatives taken to improve the situation. There has indeed been a very positive development during the past years, and a discussion of how this has happened is important. The focus is on the authors' own initiatives. This is natural, and I think many readers are unaware of wonderful resources such as the DSHB, Neuromab and NABOR and YCharOS.Personally, I would prefer a more in-depth analysis of the current problems and more specific suggestions for how to solve them.

We actually agree that a more in-depth analysis of the problems and solutions would be valuable. However, the number and complexities of the issues forces us to limit the depth attainable in a review that itself is limited in length. Several of the topics covered have been the subject of articles that focus on specific issues and are covered in more depth, and we cite these whenever possible, to allow readers to find greater depth of discussion of specific issues. One goal of the article was to emphasize the need for community efforts to work together to move towards solutions for the problems so it did not seem prudent to get too detailed in telling each group what they should do ahead of time.

[1.1] Are there really 6 million antibodies on the market. How many have ever been sold or used in publications? As for the estimated financial loss of up to $1.8bn in the US per year: how confident are the authors of this estimate? People in the business have told me that most of their revenue comes from sales of a few top-selling products: it would be good if some of the authors who are based in industry could comment on this.

Yes, according to industry sources there are about 6 million antibodies on the market. CiteAb is an online source that tracks antibodies and other reagents and they report that their database currently covers 7.4 million antibodies, though this number is clouded by the (unknown) numbers of instances in which different companies are selling the same antibody, typically with different identifiers. The Antibody Registry lists around 3 million antibodies. The Antibody Registry listings largely miss companies that are newly created and those that are in the business of aggregating and reselling antibodies from other companies; e.g., those that are targeting sales in a country specific manner, thus this might be a lower boundary. These numbers are also growing steadily in both databases.

As to the financial losses in the US, estimates have varied and we acknowledge that they are just that, estimates. We have now included more citations and include a range of estimates from those publications, rather than just the upper end that we included previously. Note that the estimates are larger in the more recent publications. Even the lowest estimate should be concerning to all readers.

It is true that most revenue from sales of antibodies comes from a relatively few top selling products. These can be found on CiteAb, along with other metrics. Reviewer comments have helped us identify CiteAb as a valuable resource that we now include in several places in the article (see reviewer comment and response below).

[1.2] The authors suggest that it is crucial to test all 6 million antibodies in multiple applications. I doubt that anyone thinks this is realistic or necessary. I would encourage them to come up with ideas for a more realistic strategy. There has to be a pre-selection step. What will it be and who will select? The next questions are who will test and how should the results be made available? My personal opinion is that money should be spent on making KO cells for as many genes as possible starting with the most widely studied proteins and make the cells available to all.

We are sorry if we left the impression that we are arguing that all commercial antibodies should be tested. Rather, we think an achievable, though still distant, goal is to identify and characterize antibodies targeting each protein in proteomes of at least the major model organisms (human, mouse, rat, worms, flies, yeast), focusing first on humans. So rather than pick an antibody and then characterize it, the approach needs to be to identify a protein and then test all available, unique antibodies (a difficult task as many resellers offer the same product under different catalog numbers, clone numbers and may even crop and change contrast on characterization data) and make the data publicly available (YCharOs approach), and in the process eliminating the bad ones and updating available information to allow researchers to make the most informed choices in purchasing their reagents. Those proteins which lack good antibodies can then be targets for antibody discovery.

In response to this comment, we have also added some additional, specific and realistic strategies that can be undertaken or expanded upon in the near future. Among these, we have expanded upon the idea of small groups of experts in a field working together, under the ‘Research scientists/end users’ section.

There already is a pre-selection step in the antibody characterization work currently underway; selection is dictated by available funding. Federal funding agencies are currently resistant to supporting such efforts broadly (though the article points out previous supported projects and highlights why many have fallen short of expectations) and this appears unlikely to change in the near future. This is also why it is important to highlight support from other funding sources. To further emphasize this point we have added some text and a reference (Davies, et al. (2013) Biochem J) that gives a great example of what can be achieved with such support. Given the lack of interest by NIH leadership in funding this work there seems little reason to get into the different ways one might prioritize pre-selection steps, although it is already being done by foundations that support such efforts.

We also strongly agree on the value of KO cells/tissues/model organisms in this arena and of making them readily available. There is currently no resource we could find for depositing and sharing KO cell lines (though some model organisms (mice, yeast, worms, flies) do have the capability to deposit and share mutants). This is an issue worthy of additional discussion and we have included text pointing out the lack of this potentially valuable resource in the article.

[1.3] Until now, the industry standard for primary screening has largely been western blotting. The implication is that we have discarded antibodies that bind to conformational epitopes. With KO cells, we can look beyond WB. How will that impact antibody development? The authors explain that there is a clear difference between the work involved in generating optimal antibody reagents and in characterizing them. I would argue that the two are closely linked and that you get what you screen for. I therefore find it interesting that Neuromab uses ELISA for the primary screen. I am sure that a lot of readers would be interested in learning more about their pipeline.

We completely agree with the issues raised in this comment; e.g., that how screening is done can influence the outcome and bias antibodies that work in certain assays over others. It is not clear how the availability of KO cell lines will alter antibody development but we emphasize throughout the article their value in characterization efforts. The issue of relying on ELISA as a first screen is a complicated one as there is not always good correlation between ELISA positive antibodies and ones that work well in other assays (as pointed out in the cited article from Gong, et al), though the fact that it is rapid, scalable, and relatively cheap means it will likely remain the industry standard at least for now. In response to the request for more details on the NeuroMab screening we have increased details in this section. We have also discussed briefly the issue of ELISA vs other assays as well as antibody production involving animals vs displays. Note that we do not go into details or comparisons of ways to generate and screen for antibodies but try to remain focused on the complicated enough issues surrounding their characterization.

[1.4] The topic of webportals is important and deserves attention. A successful portal can have immense market power. The authors mention the Antibodypedia (from the Human Protein Atlas), the Antibody Registry, the Euromabnet and YCharOS. The initiatives are great, but to me it looks as if CiteAb has already established a position as the analogue of Clarivate Analytics for antibodies. The authors may disagree, and the landscape can change quickly in the era of ChatGPT. Still, I don't think it is wise to ignore CiteAb's role in the current market.

Point well taken. We have reached out to CiteAb to get information on their activities and history and have now included a paragraph describing that work. It was also sent to the leadership at CiteAb to ensure accuracy and as a result of their efforts to write/edit and ensure the accuracy of our write up, a representative from CiteAb has been included as a co-author. We note that while a potentially useful resource they use data mining from references predominantly so risk exacerbating the problem of using numbers of references as evidence of antibody characterization. As a result we have also included citations to a couple of their online editorials, including ones that do a good job of elaborating on this issue as well as one walking the reader through steps proposed to find a good antibody.

Reviewer #2:SummaryThis is an important review that comprehensively discusses approaches to enhancing the reproducibility of antibody characterization. The authors are the key researchers working in this area from researchers to funding bodies and foundations to antibody vendors.The authors highlight the huge amounts of resources that are lost each year and the resulting poor science that is undertaken leading to unreproducible results from using antibodies that are poorly characterised and have not been sufficiently validated. The study provides a very good historical context, discussing many previous past efforts and more recent efforts to improve the rigour with which antibodies are characterised. These include the Human Protein Atlas, the Neuromab Facility, The Protein Capture Reagent Programme (PCRP), Affinomics, an EU-funded programme. The authors also discuss in depth the ‘Research Resource Identifier’ (RRID) Programme that provides a unique identifier to each antibody.The authors also highlight the Developmental Studies Hybridoma Bank (DSHB) that has played such an important role in distributing hybridoma clones and more recently recombinant antibody cDNA vectors.

We want to clarify that DSHB distributes recombinant and monoclonal antibodies but not vectors that may be used to express them. We have gone over the text to ensure this point is clear.

The authors also discussed the Clinical Proteome Tumour Analysis Consortium (CPTAC) that provides antibodies for cancer research. The authors also discuss a very valuable effort, started by the Structural Genome Consortium, called the Antibody Characterisation Through Open Science (YCharOS). This consortium have rigorously tested over 1000 antibodies and characterised detailed antibody reports on many of these, and have found that a significant proportion of all antibodies that have been used in many previous publications shockingly do not detect endogenous proteins.The authors also discuss a community called Only Good Antibodies (OGA) that are also attempting to validate antibodies in a consistent manner and provide this information to the research community. The authors also highlight the important roles that researchers, vendors, universities, journals, funding society and disease foundations can have to mandate that researchers only use well characterised antibodies, or if the antibody hasn't been well characterised previously, the authors need to undertake essential characterisation, demonstrating that the antibodies they are using are indeed recognising the designated target, preferably using a knockout or knockdown approach. The authors also highlight important work undertaken by the Michael J. Fox Foundation for Parkinson's research where they've developed a unique "research tools programme" which focuses on characterisation and validation of antibodies used for research. Thus far, 200 research tools have been produced to date that are all highly characterised, and groups using these reagents can be reassured that the antibodies are very well characterised.The authors also discuss the importance of trying to move away from polyclonal antibodies to recombinant antibodies in which the sequences are known and antibodies can be expressed more consistently in human cell lines. Ideally having sequences available for antibodies certainly helps researchers to ensure that the antibody is the correct reagent and enables them to express these reagents as scale in their laboratories, rather than having to purchase these critical reagemts at high cost from a vendor.The authors provide a lot of important suggestions and recommendations including improving awareness and training in laboratories of the importance of rigorous antibody characterization has in ensuring the reproducibility of life sciences research.This article is a must read for Life Scientists at any stages of their careers who make use of antibodies for their research. I recommend acceptance of this study.[2.1] The authors highlight the major problem with companies that distribute antibodies to multiple other companies resulting in the same antibodies having different RRID numbers, that causes major confusion. It would be important to put in place a system whereby if a company purchases an antibody from another vendor that they tried to ensure that the original RRID number is used for this purpose-this may be difficult to achieve. The authors should comment on this.

Linking antibodies with their sequences would also be good but also unlikely to work as most antibody companies do not disclose the sequences of their antibodies for commercial reasons. It would be good if the authors could also comment on this.

Both great suggestions/comments. There are ongoing discussions among stakeholders to develop a better system in which each antibody is given one and only one RRID number. Those involved admit this will be difficult to enforce and track but all agree it is a worthwhile effort. We have included this in the article as the worthy goal that it is.

Under the RRID section we have added:

“It is also important to note that one antibody can be sold by multiple different vendors with each using a different RRID number, and different lots of the same manufacturer’s antibody will have the same RRID, even if there may be significant lot to lot variation. To the extent possible, this practice should be replaced by better practices, including assigning one RRID to a reagent so that each antibody is given one and only one RRID number.”

Under Antibody vendors and repositories we have added this:

“Vendors should also take the lead in ensuring that each antibody is assigned one, and only one, RRID number to allow better tracking and linkage to characterization data. When distributing to other vendors they should have the opportunity to make this a requirement.”

As to the linkage of sequences to antibodies, you are correct that vendors will not do so. This is in response to experience in which other companies, typically overseas, have used the sequence information to generate the same product and sell it without compensation to the originator. While a worthy goal, we believe it is unrealistic today, though now include the following statement in the article, under the NeuroMab section:

“The mAbs and hybridomas that produce them are distributed through the DSHB and the rAb sequences and plasmids are available at Addgene. It is worth noting here both the value in making sequences of antibodies available to researchers but also the limitations on vendors, as commercial enterprises, who cannot readily do so without risking use of the information by competitors.”

[2.2] YCharOS found that on average 12 publications per protein target contain data from an antibody that failed to recognise the endogenous targets. YCharOS is an extremely important effort and I hope that it will be able to continue expanding their efforts to properly validate many of the widely used antibodies. This effort represents a drop in the ocean and the feasibility of dramatically scalling up the efforts of YCharOS to test a greater proportion of all widely used antibodies seems very challenging.It would be good if the authors could say more about the challenges involved in scaling up such efforts.

As with so many things in research, the challenges in scale up get back to funding. This is a theme we try to bring out throughout the article and hope that federal funding agencies are paying attention. YCharOS is in the process of scaling up, with sites in Canada and the UK currently and a third planned in the US. We also believe that groups of experts in each field are ideal for shared efforts in this arena and may allow for funding in more focused projects, targeting perhaps all members of a protein family or the known and emerging players in a specific pathology or pathway. The practical limitation of scaling up efforts is that among the most common assays used by researchers (WB and IHC) are not readily scalable or automated without migrating to a platform that may not sufficiently resemble what is done in the average research laboratory.

[2.3] It would be helpful to have a table or figure outlining the key steps/criteria that the authors believe are needed to thoroughly characterise antibodies for immunoblotting, immunofluorescence, immunoprecipitation, immunohistochemistry, and flow cytometry. This might help students and postdocs plan their experiments to ensure that the data required to validate the antibodies that they're using is available. Ideally, vendors would also sign up to this level of characterisation data being provided in the antibody datasheets. Ideally third-party companies or consortia could be available to undertake this important QC work in the future and feed the back to the community.

We agree and found this to be an excellent suggestion that we are happy to comply with. We generated a new Figure (Figure 1) that summarizes these issues and hope it helps clarify in the minds of students and all end users the issues and steps that can be taken.

In addition, YCharOS has recently submitted a manuscript, currently under review at a leading Methods journal, with step-by-step protocols for WB/IP/IF and put it up on Protocols Exchange, a preprint server. Representatives from 10 leading antibody manufacturers are co-authors on this article, demonstrating the adoption of these protocols by the industry. Also, NeuroMab has made their IHC protocols openly available.

For reasons included in the article, it remains financially unsupportable for vendors to perform the characterization work comparable to YCharOS protocols as costs for analyses of commercial antibodies targeting one protein may be estimated at ~$30,000, which is far more than a company makes from selling the average antibody.

[2.4] If the sequences of the recombinant antibody are known, using Alphafold 3 it should be possible to predict the structure of the antibody antigen complex. This might enable researchers to generate subtle point mutations in the recombinant antibody that ablates their affinity for the intended target. Such mutated antibodies could serve as useful control reagents for certain applications such as IF, IHC, flow cytometry, IP, to ensure that the signals that are observed are indeed mediated by binding to the antigen and not to a non-specific site.It would be good if the authors could comment on this (including the possibility that vendors might start providing these control reagents in the future).

This too is an excellent point and we have included the future potential for deep learning algorithms like AlphaFold to play in this field, in the section headed Technology Development. We also used this as another argument in support of vendors making sequence data readily available, though for reasons already given this is unlikely to become standard practice in the industry.

The newly added text:

“Finally, a future role for technology in this field that will likely play increasingly important roles in the development and optimization of antibodies will come from deep learning models, such as AlphaFold. These models should enable predictions of the antibody-antigen complex, aid in the identification of the epitope targeted, and help determine if folding, post-translational modifications or other issues may influence the output from use of the antibody. These future prospects are another reason to encourage open access to antibody sequences of rAbs.”

The use of AlphaFold to generate point mutations for use as negative controls was not deemed as useful for a number of reasons including: (1) modified antibodies could develop specificity for other unwanted proteins and (2) the use of WT versus modified antibodies won’t confirm whether the signal from the WT antibody was coming from the intended target protein or unwanted proteins.

[2.5] An important recommendation that is not made in the review (perhaps because vendors are co-authors) is that vendors should sign up to withdraw all antibodies found not to work or detect the endogenous target by trusted third party testers or consortia. Again it would be good to discuss this issue in the review.

Responsible vendors have already stepped up and are taking appropriate steps in response to data becoming available. This is best demonstrated by the fact that the YCharOS manufacturer partners have removed 20% of antibodies tested by YCharOS that failed in testing, and 40% of antibodies have had their recommendations changed (Ayoubi et al., *eLife*, 2023). We also point out again that just because an antibody does not perform well in one assay does not mean it won’t be useful in a different use or in the same assay but under different conditions. So it is not reasonable to insist on withdrawal of any antibody, except those found to not bind the stated target.

In summary, we again thank the reviewers for their input and believe we have responded appropriately to each of their comments/suggestions and as a result believe the manuscript should be ready for publication.